# Adherence to COVID-19 preventive practice and associated factors among pregnant women in Gondar city, northwest Ethiopia, 2021: Community-based cross-sectional study

Wubedle Zelalem Temesgan[1]*, Mastewal Belayneh Aklil[1], Henok Solomon Yacob[2], Esubalew Tsega Mekonnen[2], Elias Derso Tegegne[2], Esubalew Binega Tesfa[2], Eshetie Melkie Melese[2], Tewodros Seyoum[3]

1 Department of Clinical Midwifery, School of Midwifery, College of Medicine and Health Sciences, University of Gondar, Gondar, Ethiopia, 2 Department of General Midwifery, School of Midwifery, College of Medicine and Health Sciences, University of Gondar, Gondar, Ethiopia, 3 School of Midwifery, College of Medicine and Health Sciences, University of Gondar, Gondar, Ethiopia

* wubedlezelalem@gmail.com

## Abstract

### Background

Coronavirus disease 2019 (COVID-19) causes more than five million deaths worldwide. Pregnant women are at high risk for infection due to the physiologic change in the immune and cardiopulmonary system and also it increases the risk of severe disease, intensive care unit admission, and receive mechanical ventilation when compared with non-pregnant women. It is associated with adverse maternal and neonatal outcomes. So pregnant women need to have adhered to preventive measures to prevent COVID-19 related consequences. Therefore, this study aimed to assess adherence toCOVID-19 preventive practice and associated factors among pregnant women in Gondar city, northwest Ethiopia.

### Methods

A community-based cross-sectional study was conducted from July 1st to 30th, 2021, in Gondar city. A cluster sampling technique was employed to select 678 pregnant women. Data were collected using a pre-tested, face-to-face interviewer-administered questionnaire. Data were entered into EPI DATA version 4.6 and exported to SPSS version 25 for analysis. Both bivariable and multivariable logistic regression analysis was fitted to identify associated factors. Adjusted odds ratio with a 95% confidence interval was used to report the association between covariates and the outcome variable.

### Results

The prevalence of good adherence to COVID-19 preventive practice was 44.8% (95% CI: 41.3, 48.7). Maternal age (≤24 years) [AOR = 2.89, 95% CI: 1.37, 6.10], maternal education (secondary school) [AOR = 2.95, 95% CI: 1.58, 5.53] and (college and above) [AOR = 4.57,95% CI: 2.42, 8.62], having ANC follow up [AOR = 2.95, 95% CI: 1.35, 6.46] and

**Data Availability Statement:** All relevant data are within the manuscript and its Supporting information files.

**Funding:** The authors received no specific funding for this work.

**Competing interests:** The authors declared no competing interest.

**Abbreviations:** ANC, Antenatal Care; AOR, Adjusted Odds Ratio; CI, Confidence Interval; COR, Crude Odds Ratio; COVID-19, Corona virus disease 2019; SARS-CoV-2, Severe Acute Respiratory Syndrome Coronavirus 2; WHO, World Health Organization.

adequate knowledge towards COVID-19 [AOR = 1.70, 95% CI: 1.20, 2.41] were significantly associated with good adherence to COVID-19 preventive practice.

## Conclusion

In this study, adherence towards COVID-19 preventive practice in pregnant women is low. Hence, it is important to strengthen women's awareness about COVID-19 through different media and health education. In addition, empowering women to attain ANC and special consideration should be given to women who had no formal education.

## Introduction

Coronavirus disease 2019 (COVID-19) is a novel virus that is caused by severe acute respiratory syndrome coronavirus 2 (SARS-CoV-2) first detected at the end of December 2019, in Wuhan city China [1, 2]. As COVID-19 rapidly distribute throughout the world, World Health Organization (WHO) declared it as a public health emergency of international concern on January 30, 2020 [3]. Following this, due to the continuous rise of COVID-19 related cases and deaths, WHO declared the disease a global pandemic on, March 11, 2020 [4–6]. The Ethiopian government announced the first case was detected on March 13, 2020 [7]. Currently, from the total population of Ethiopia (116.4 million), more than four million people screened for COVID-19 and more than six thousand deaths are recorded [8, 9].

Globally, COVID-19 had spread to over 219 countries and has influenced different devastating problems to the human race including mortality and morbidity, and social and economic crisis [7, 10]. As November 20, 2021 report, confirmed COVID-19 cases were 257,123,909, deaths 5,158,990 and 232,181,236 cases were recovered from COVID-19 in the world, and 8,654,048 cases, 221,941 deaths and 8,035,451 cases were recovered from COVID-19 in Africa and 369,867 cases, 6,662 deaths and 346,547 recovery were recorded in Ethiopia [11]. The presence of different COVID-19 variants (Delta, Kappa, Alpha, Beta etc.), with high mutation characteristics and ineffectiveness of the vaccine makes the pandemic difficult to control in the world. This virus variation possibly creates more worst complications in the third wave of the pandemic [12].

World Health Organization (WHO) has recommended several preventive measures to curve down the spread of the disease like regular handwashing with water and soap, social distancing, wearing mask, covering the mouth while coughing and sneezing, avoiding touching eyes, nose, and mouth [13]. In addition, scholars showed that consumption of fiber rich foods and micronutrients can boost immunity and prevent adverse pregnancy outcomes in COVID-19 infected women [14, 15]. The government of Ethiopia has also engaged in different activities to halt the spread of the virus in the country like Community mobilization, public awareness creation, isolation, compulsory quarantine, and treatment, strict passenger screening, house to house screening, and scaling up diagnostic tests and treatment centers [16].

Every person is at risk of becoming infected with COVID-19. However, pregnant women are at high risk for infection because the physiologic change in the immune and cardiopulmonary system during pregnancy makes them more susceptible to SARS-CoV-2 infection when compared to the general population [17, 18]. Centre for Disease Control and Prevention (CDC) report showed that pregnant women with COVID-19 infection are at increased risk of severe disease, intensive care unit admission, and receiving mechanical ventilation when compared to non-pregnant patients [17, 19, 20]. Moreover, it is associated with maternal and

neonatal complications like miscarriage, preterm births, intrauterine growth restriction, fetal distress, and preeclampsia [17, 21, 22]. A study showed that COVID -19 can pass directly from mother to fetus during pregnancy [23]. There is also evidence that revealed that during the pandemic the risk of perinatal anxiety, depression, and domestic violence also increase in pregnant women [24]. Even though pregnant women have not been included in any COVID-19 vaccine clinical trials, the American College of Obstetricians and Gynecologists and the Society for Maternal-Fetal Medicine recommend that pregnant women can take COVID-19 vaccines if they prefer to be vaccinated [25–27].

Although the spread of COVID -19 increases every day in Ethiopia, the vaccine is not accessible for pregnant women. so pregnant women need to have adhered to preventive measures set by the Ministry of Health (MoH) to decrease the likelihood of being infected by the virus. This reduces maternal and neonatal morbidity and mortality. In Ethiopia, researches addressing the highly vulnerable pregnant women are scarce, particularly in the study area. Therefore, this study aimed to assess adherence with COVID-19 preventive practice and associated factor among pregnant women in Gondar city, northwest Ethiopia.

## Methods and materials

### Study design and setting

A community-based cross-sectional study was conducted in Gondar city from July $1^{st}$ to $30^{th}$, 2021. Gondar city is located 166 km far from Bahirdar, the capital city of Amhara regional state, and 750 km northwest of Addis Ababa, the capital city of Ethiopia. According to the Population projection of Ethiopia for all regions at the woreda level from 2014–2017, the total population of the town was estimated to be 306,246. Among these, 156,276 of the population are females [28]. Currently, it has 1 governmental comprehensive specialized hospital, 8 governmental health centers, 22 health posts, 1 private primary hospital, and 1 general hospital serving the population of the city and catchment area.

### Study population and eligibility criteria

The study population were all pregnant women in the selected clusters of Gondar city during the data collection period and residing in the study area for at least six months before the data collection period.

### Sample size determination and sampling procedure

The sample size was determined by using a single population proportion formula by considering a proportion of good practice 47.6% [29], level of confidence 95%, margin of error 5%. Thus, the sample size $(n) = \frac{(Z\alpha/2)^2 p(1-p)}{d^2} = \frac{(1.96)^2 0.476(1-0.476)}{(0.05)^2} = 384$. After considering a design effect of 1.5, and a non-response rate of 10%, the total sample size was 634. Gondar city has 22 kebeles (the smallest administrative unit in Ethiopia) and five kebeles were selected by a lottery method. All eligible participants in the selected cluster were included in the study, making the final sample size of 678.

### Variables of the study

Adherence with COVID-19 preventive practice was the outcome variable whereas, maternal age, religion, maternal educational status, maternal occupation, marital status, husband educational status, husband occupation, number of families living together, average monthly income, gravidity, number of alive children, having ANC visit, number of ANC visit, condition

of pregnancy, history of previous adverse pregnancy outcome, history of chronic illness and knowledge about COVID- 19 preventive practice were independent variable of the study.

## Operational definitions and measurements

**Adherence to COVID-19 preventive practice.** It was measured by using 6 item questions derived from WHO recommendation on prevention measures against COVID -19 and participants were asked to answer their practice in the last 14 days. Each correct response weighed 1 point and 0 for an incorrect response. Participants who scored 100% on the practice of COVID-19 preventive measures were considered as having good adherence to COVID- 19 preventive practice. Whereas, participants who scored < 100% were considered as having poor adherence to COVID- 19 preventive practice [30, 31].

**Knowledge.** It was measured by using 14 item questions that include clinical presentation, risk factors, prognosis, and prevention measures of COVID-19. Each correct response weighed 1 point and 0 for an incorrect response. Participants who scored ≥65% of knowledge questions were considered as having adequate knowledge and those who scored < 65% were considered as having inadequate knowledge [31].

## Data collection tools & procedures

The data collection tool was developed by reviewing related literature [29, 31–33]. Data were collected using semi-structured, pretested questionnaires through a face-to-face interview. The questionnaire contains socio-demographic characteristics, obstetric-related variables, and knowledge and preventive practice of COVID-19 related questions. Five BSc midwives and one MSc clinical midwife were recruited for data collection and supervision respectively.

## Data quality control measures

The questionnaire was prepared in English and then translated into the local language Amharic with the assistance of language experts then translated back to English to maintain consistency of the tool. Before the actual data collection, the questionnaire was pretested on 5% of the total sample size at Kola Diba district to check the response, language clarity, and appropriateness of the questionnaire. One-day training was given for data collectors and supervisor about the aim of the study, contents of the tool, and techniques of data collection. In addition, the data collectors and supervisor were informed regarding important precautions to be taken to prevent COVID-19 infection. During data collection, data collectors were supervised for any difficulties. The consistency and completeness of the data were checked by the data collectors and supervisor and the incomplete data were discarded before data entry.

## Data processing & analysis

The data were checked, coded, and entered into Epi data version 4.6 and then exported to SPSS version 25 for analysis. Descriptive statistics like frequencies, percentages, mean and standard deviation were computed. Bivariable logistic regression was done to identify the association between independent and dependent variables, variables with p-values less than 0.2 were entered into multivariable logistic regression analysis. Adjusted odds ratio with 95% CI was used to determine the level of significance at p-value of ≤ 0.05.

## Ethical consideration

Ethical clearance was obtained from the University of Gondar ethical review committee. A supportive letter was obtained from selected kebeles administrates. Written informed consent

was taken from each study participant after a clear explanation of the aim of the study. Study participants were also informed that they had the full right to withdraw from the interview at any time.

## Results

### Socio-demographic characteristics of study participants

In this study, a total of 663 women were included in the analysis making a response rate of 97.8%. The age of the participants ranges from19 to 44 years with a mean age of 31.25 (±5.525 SD) years. More than half (57.5%) of the respondents were in the age group of 25–34 years. Nearly two-thirds (66.4%) of study participants were Orthodox Christians in religion. The majority (94%) of the study participants were married. Regarding husband's educational status nearly one-third (31.5%) of their husband had attended secondary school (Table 1).

### Obstetric and reproductive characteristics of study participants

Among the total study participants, more than three-fourths (79.2%) and the majority (93.4%) of them were multigravida in their gravidity and had antenatal care (ANC) follow up respectively. Regarding the condition of pregnancy 601 (90.6%) of the pregnancy was planned (Table 2).

### Knowledge about COVID-19

From the total study participants, 422 (63.7%) had adequate knowledge about COVID-19. Almost all respondents heard about COVID-19. More than half (52.3%) and (53.5%) of pregnant women said that sore throat and runny nose are a symptom of COVID-19 respectively. Five hundred twenty-seven (79.48%) respondents know that pregnant women have higher risk for COVID-19 than other populations (Table 3).

### Adherence to COVID-19 preventive practice

The finding of this study showed that only 44.8% (95% CI: 41.3, 48.7) of the participants had good adherence to COVID-19 preventive practice (Fig 1). About three-fourths (74.1%) and majority (90.3%) of participants wash hands with water and soap or with alcohol-based sanitizer and wear a face mask in public to decrease the spread of infection respectively (Fig 2).

**Factors affecting adherence to COVID-19 preventive practice.**   In the bi-variable analysis: maternal age, maternal educational status, marital status, number of families living together, average monthly income, gravidity, number of alive children, ANC follow up and knowledge about COVID-19 were significantly associated with COVID-19 preventive practice adherence at a p-value <0.2. Of these variables: maternal age, maternal educational status, ANC follow up and knowledge about COVID-19 were statistically significant to adherence with COVID-19 preventive practice in multivariable logistic regression.

Study participants whose age ≤ 24 years were 2.89 times more likely to have good adherence to COVID-19 preventive practice as compared to participants whose age group of ≥35 years [AOR = 2.89, 95% CI: 1.37, 6.10]. Respondents who had an educational level of secondary school and college and above were 2.95, and 4.57 times more likely to have good adherence to COVID-19 preventive practice as compared to women who can not read and write respectively [AOR = 2.95, 95% CI: 1.58, 5.53], [AOR = 4.57, 95% CI: 2.42, 8.62]. The odds of good adherence with COVID-19 preventive practice were 2.95 times higher among women who had ANC follow-up as compared to their counterparts [AOR = 2.95, 95% CI: 1.35, 6.46]. Pregnant women who had adequate knowledge about COVID-19 were 1.7 times more likely to

**Table 1. Sociodemographic characteristics of pregnant women in Gondar city, northwest Ethiopia 2021 (n = 663).**

| Variables | Frequency | Percent (%) |
|---|---|---|
| **Maternal age** | | |
| ≤24 years | 43 | 6.5 |
| 25–34 years | 381 | 57.5 |
| ≥35 years | 239 | 36.0 |
| **Religion** | | |
| Orthodox | 440 | 66.4 |
| Muslim | 178 | 26.8 |
| Protestant | 41 | 6.2 |
| Catholic | 4 | 0.6 |
| **Maternal educational status** | | |
| Unable to read and write | 71 | 10.7 |
| Able to read and write | 160 | 24.1 |
| Elementary(1–8) | 106 | 16.0 |
| Secondary School(9–12) | 167 | 25.2 |
| College and above | 159 | 24.0 |
| **Maternal occupation** | | |
| Housewife | 273 | 41.2 |
| Civil servant | 170 | 25.6 |
| Private Business | 186 | 28.1 |
| Farmer | 14 | 2.1 |
| Others* | 20 | 3.0 |
| **Marital status** | | |
| Married | 623 | 94.0 |
| Unmarried | 40 | 6.0 |
| **Husband educational status (n = 623)** | | |
| Unable to read and write | 33 | 5.3 |
| Able to read and write | 174 | 27.9 |
| Elementary(1–8) | 51 | 8.2 |
| Secondary School(9–12) | 196 | 31.5 |
| College and above | 169 | 27.1 |
| **Husband occupation (n = 623)** | | |
| Civil servant | 209 | 33.5 |
| Private Business | 352 | 56.5 |
| Farmer | 56 | 9.0 |
| Others** | 6 | 1.0 |
| **Number of family living together** | | |
| ≤2 | 106 | 16.0 |
| 3–4 | 300 | 45.2 |
| ≥5 | 257 | 38.8 |
| **Average monthly income** | | |
| <5000 ETB | 177 | 26.7 |
| 5000–10000 ETB | 323 | 48.7 |
| >10000 ETB | 163 | 24.6 |

Others*; daily laborer and student, Others**; daily laborer and unemployed, ETB; Ethiopian birr

**Table 2. Obstetric and reproductive characteristics of pregnant women in Gondar city, northwest Ethiopia, 2021 (n = 663).**

| Variables | Frequency | Percent (%) |
|---|---|---|
| **Gravidity** | | |
| Primigravida | 134 | 20.8 |
| Multigravida | 529 | 79.2 |
| **Number of alive children** | | |
| No alive children | 141 | 21.3 |
| <3 | 345 | 52.0 |
| ≥3 | 177 | 26.7 |
| **ANC follow up** | | |
| Yes | 619 | 93.4 |
| No | 44 | 6.6 |
| **Number of ANC follow up (n = 619)** | | |
| ≤3 | 444 | 71.7 |
| >3 | 175 | 28.3 |
| **Condition of pregnancy** | | |
| Planned | 601 | 90.6 |
| Unplanned | 62 | 9.4 |
| **History of previous adverse pregnancy outcome** | | |
| Yes | 121 | 18.3 |
| No | 542 | 81.7 |
| **History of chronic illness** | | |
| Yes | 132 | 19.9 |
| No | 531 | 80.1 |

**Table 3. Knowledge of pregnant women about COVID-19 in Gondar city, northwest Ethiopia 2021 (n = 663).**

| Knowledge item | Correct response n (%) | Incorrect response n (%) |
|---|---|---|
| Ever heard about COVID-19 | 660 (99.5) | 3 (0.5) |
| COVID-19 is viral disease | 561 (84.6) | 102 (15.4) |
| Respiratory droplets and close contact are the main transmission route. | 578 (87.2) | 85 (12.8) |
| All people are generally susceptible to COVID-19 | 459 (69.2) | 204 (30.8) |
| Dry cough is a symptom of COVID-19 | 521 (78.6) | 142 (21.4) |
| Fever is a symptom of COVID-19 | 405 (61.1) | 258 (38.9) |
| Headache is a symptom of COVID-19 | 502 (75.7) | 161 (24.3) |
| Sore throat is a symptom of COVID-19 | 347 (52.3) | 316 (47.7) |
| Runny nose is a symptom of COVID-19 | 355 (53.5) | 308 (46.5) |
| Difficulty of breathing is a symptom of COVID-19 | 557 (84.0) | 106 (16.0) |
| Stay at home and wearing face mask can prevent COVID-19 | 630 (95.0) | 33 (5.0) |
| People with co-existing disease has poor prognostic outcome | 427 (64.4) | 236 (35.6) |
| Person with COVID-19 can transmit the virus to others without developing sign | 437 (65.9) | 226 (34.1) |
| Pregnant women are at high risk than others | 527 (79.5) | 136 (20.5) |

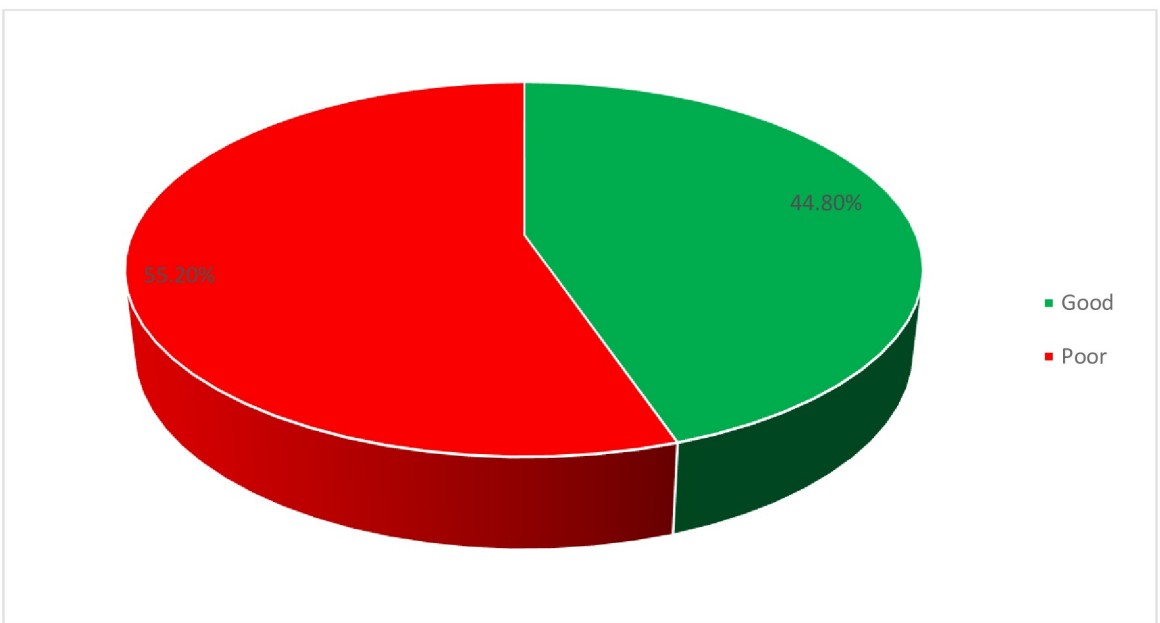

**Fig 1. Adherence with COVID-19 preventive practice among pregnant women in Gondar city, northwest Ethiopia, 2021.**

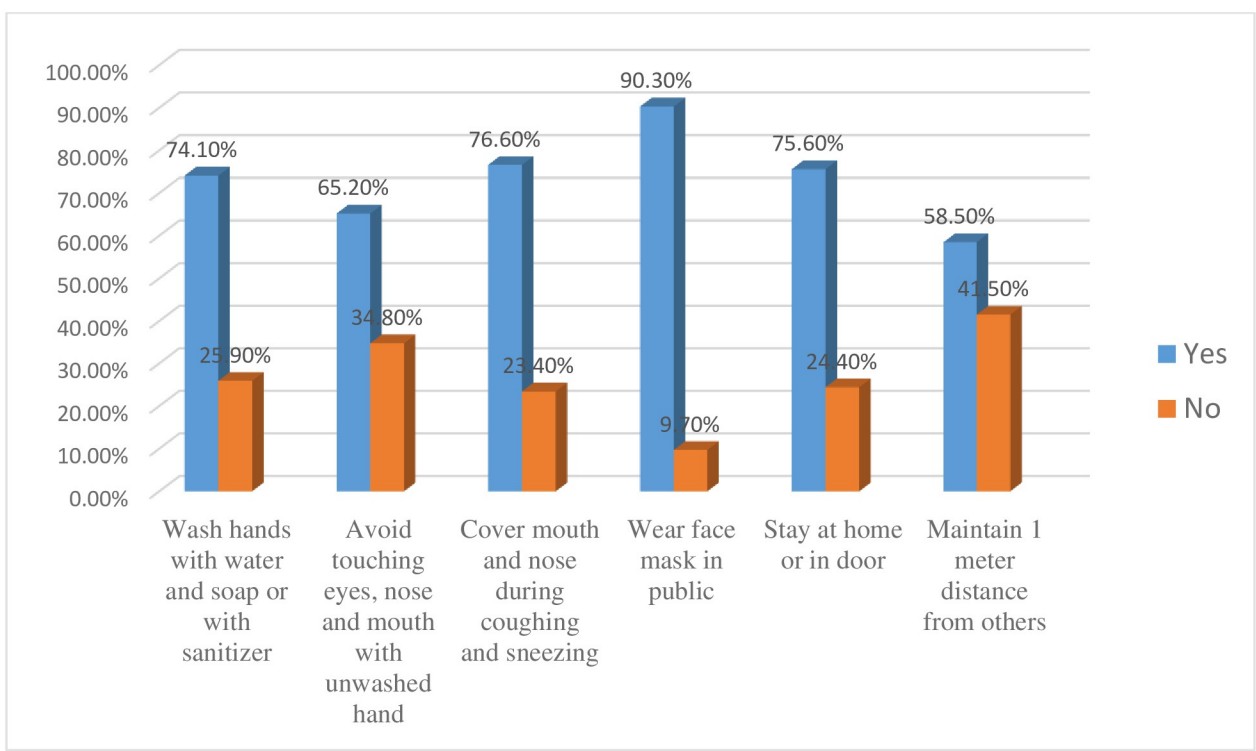

**Fig 2. Adherence to specific COVID-19 preventive practice among pregnant women in Gondar city, northwest Ethiopia, 2021.**

**Table 4. Bivariable and multivariable logistic regression analysis of factors affecting adherence to COVID 19 preventive practice among pregnant women in Gondar city, northwest Ethiopia, 2021 (n = 663).**

| Variables | Adherence | | COR (95%CI) | AOR (95% CI) |
|---|---|---|---|---|
| | Good | poor | | |
| **Maternal age** | | | | |
| ≤24 | 28 | 15 | 2.73 (1.39, 5.38) | **2.89 (1.37, 6.10)**[*] |
| 25–34 | 172 | 209 | 1.20 (0.87, 1.67) | 1.05 (0.74, 1.52) |
| ≥35 | 97 | 142 | 1 | 1 |
| **Maternal educational status** | | | | |
| Unable to read and write | 19 | 52 | 1 | 1 |
| Able to read and write | 45 | 115 | 1.07 (0.57, 2.01) | 0.93 (0.53, 1.89) |
| Elementary(1–8) | 38 | 68 | 1.53 (0.79, 2.95) | 1.36 (0.68, 2.71) |
| Secondary School(9–12) | 91 | 76 | 3.28 (1.78, 6.01) | **2.95 (1.58, 5.53)**[*] |
| College and above | 104 | 55 | 5.17 (2.79, 9.61) | **4.57 (2.42, 8.62)**[**] |
| **Marital status** | | | | |
| Married | 272 | 351 | 0.46 (0.24, 0.90) | 0.56 (0.27, 1.16) |
| Unmarried | 25 | 15 | 1 | 1 |
| **Number of family living together** | | | | |
| ≤2 | 56 | 50 | 1.82 (1.15, 2.87) | 0.58 (0.24, 1.39) |
| 3–4 | 143 | 157 | 1.48 (1.05, 2.07) | 0.76 (0.45, 1.30) |
| ≥5 | 98 | 159 | 1 | 1 |
| **Average monthly income** | | | | |
| <5000 | 67 | 110 | 1 | 1 |
| 5000–10000 | 142 | 181 | 1.29 (0.88, 1.87) | 0.98 (0.65, 1.48) |
| >10000 | 88 | 75 | 1.93 (1.25, 2.97) | 1.48 (0.91, 2.41) |
| **Gravidity** | | | | |
| Primigravida | 74 | 60 | 1.69 (1.15, 2.48) | 1.42 (0.23, 8.85) |
| Multigravida | 223 | 306 | 1 | 1 |
| **Number of alive children** | | | | |
| No alive children | 77 | 64 | 2.35 (1.49,3.70) | 1.47 (0.79, 2.71) |
| <3 | 160 | 185 | 1.69 (1.16, 2.46) | 1.26 (0.80, 1.98) |
| ≥3 | 60 | 117 | 1 | 1 |
| **ANC follow up** | | | | |
| Yes | 287 | 332 | 2.94 (1.43, 6.05) | **2.95 (1.35, 6.46)**[*] |
| No | 10 | 34 | 1 | 1 |
| **Knowledge about COVID-19** | | | | |
| Adequate knowledge | 208 | 214 | 1.66 (1.20, 2.29) | **1.70 (1.20, 2.41)**[*] |
| Inadequate knowledge | 89 | 152 | 1 | 1 |

COR-crude odds ratio, AOR-adjusted odds ratio, CI-confidence interval, 1-reference category,

[*]p≤0.05,

[**] p <0.01

adhere to COVID-19 preventive practice than pregnant women who had inadequate knowledge about COVID-19 [AOR = 1.70, 95% CI: 1.20, 2.41] (Table 4).

## Discussion

This study has attempted to assess adherence with COVID-19 preventive practice and associated factors among pregnant women in Gondar city, northwest Ethiopia, 2021. The study

result showed that only 44.8% (95% CI: 41.3, 48.7) of the study participants had good adherence with COVID-19 preventive practice. These results are comparable with studies conducted in Ghana 46.6% [32], Egypt 42.86% [34], Wollega zone, Ethiopia 43.6% [33], and Debre Tabor, Ethiopia 47.6% [29].

This finding is lower than studies conducted in Nigeria 79.2% [35] and studies done in some parts of Ethiopia such as Gurage zone, 76.2% [31], Debre Berhan, 56.1% [36]. The possible reason for this discrepancy could be due to the difference in the study setting, variation in measuring outcome variable, and sociodemographic characteristics of the study participants. All the above-mentioned studies were institution-based cross-sectional studies. Women who came for ANC follow-up can get information about COVID-19 and are highly likely to practice it. Studies in Nigeria and Debre Berhan used mean score and 80% as a cut of point to say participants had good adherence respectively. Whereas, in this study, 100% was used as a cut of point to measure the outcome variable. In addition, 91% and 87.2% of study participants in Nigeria attained at least primary education and had good knowledge of COVID-19 infections respectively. However, in this study, more than one-third (34.8%) of pregnant women had no formal education and 63.7% had adequate knowledge about COVID-19.

However, the result of this study is higher than the study in Ebonyi, state, Nigeria 30.3% [37]. The possible explanation for the discrepancy might be due to differences in the time gap of the study and study setting. As time goes community awareness creation and Information dissemination strategies also improved and it might have a positive impact on preventive practice. In addition, 39.1% of study participants in Ebonyi, state were rural dwellers while this study was conducted on urban residents. It has been evidenced that pregnant women living in urban areas were highly likely to practice preventive measures [32, 33].

In this study, maternal age is one of the variables significantly associated with good adherence to COVID-19 preventive practice. Study participants whose age ≤24 years were 2.89 times more likely to have good adherence with COVID -19 preventive practice as compared to participants whose age was≥35 years. This finding was supported by a study conducted in Nigeria [35] and studies in different parts of Ethiopia, Wollega zones, [33], Gurage zone, [31]. This might be due to younger women having access to information on COVID-19 (risk, mode of transmission, and way of preventive measures) via social media and mass media. Also, it might be younger women have better educational attainment than older women. Pregnant women who had a higher level of education have good adherence with COVID -19 preventive practice. On the contrary, studies conducted in Ghana [32] and Debre Berhan, Ethiopia [36], as age increase the odd of good preventive practice also increase.

The study also revealed that women with educational level of secondary school and, college and above were 2.95 and 4.57 times more likely to have good adherence with COVID-19 preventive practice as compared to women's who cannot able to read and write respectively. This finding was supported by studies conducted in Ghana [32], Debre Tabor, Ethiopia [29]. This might be due to the fact that educated people can access information about COVID -9 including its prevention method from different sources than those unable to read and write. In addition, educated people might have a better understanding of preventive measures and consequences associated with not doing this preventive practice.

ANC follow up also one of the important predictors of good adherence to COVID-19 preventive practice. The odds of good adherence with COVID-19 preventive practice were 2.95 times higher among women who had ANC follow-up as compared to their counterparts. This might be pregnant women who had ANC follow-up have a chance to get information about the pandemic from the health care providers during their visit. Study showed that obtaining COVID-19 education at a health facility was significantly associated with adequate knowledge on COVID-19 prevention [32].

This study also showed that there was a significant association between the knowledge of the respondents and adherence to COVID-19 preventive practice. Pregnant women who had adequate knowledge about COVID-19 were 1.7 times more likely to adhere to COVID-19 preventive practice than pregnant women who had inadequate knowledge about COVID-19. This finding was supported by studies conducted in Debre Berhan [36] and Debre Tabor, Ethiopia [29]. This might be due to women who had prior knowledge about the pandemic including the benefits of preventive measures may apply it to protect themselves and their families against the pandemic.

## Limitation of the study

The cross-sectional nature of the study design might not possible to infer the cause and effect relationship between pregnant women's adherence with COVID-19 preventive practice and its associated factors. There might be also a social desirability bias.

## Conclusion

In this study, adherence with COVID-19 preventive practice among pregnant women is low. Maternal age, maternal occupational status, ANC follow up and knowledge about COVID-19 were factors affecting adherence with COVID-19 preventive practice. Therefore, it is important to strengthen the health education of pregnant women about COVID-19 preventive practice through different media. In addition, empowering women to attain ANC and special consideration should be given to women who had no formal education.

## Supporting information

**S1 Questionnaire. English version questionnaire.**
(DOCX)

**S1 Dataset.**
(SAV)

## Acknowledgments

We would like to thank the university of Gondar for Ethical approval. We would also like to extend our gratitude to each Keeble administrative, data collectors, supervisor and study participants.

## Author Contributions

**Conceptualization:** Wubedle Zelalem Temesgan.

**Data curation:** Wubedle Zelalem Temesgan, Mastewal Belayneh Aklil, Henok Solomon Yacob, Esubalew Tsega Mekonnen, Elias Derso Tegegne, Esubalew Binega Tesfa, Eshetie Melkie Melese, Tewodros Seyoum.

**Formal analysis:** Wubedle Zelalem Temesgan, Mastewal Belayneh Aklil, Henok Solomon Yacob, Esubalew Tsega Mekonnen, Elias Derso Tegegne, Esubalew Binega Tesfa, Eshetie Melkie Melese, Tewodros Seyoum.

**Funding acquisition:** Wubedle Zelalem Temesgan, Mastewal Belayneh Aklil, Henok Solomon Yacob, Esubalew Tsega Mekonnen, Elias Derso Tegegne, Esubalew Binega Tesfa, Eshetie Melkie Melese, Tewodros Seyoum.

**Investigation:** Wubedle Zelalem Temesgan.

**Methodology:** Wubedle Zelalem Temesgan, Mastewal Belayneh Aklil, Henok Solomon Yacob, Esubalew Tsega Mekonnen, Elias Derso Tegegne, Esubalew Binega Tesfa, Eshetie Melkie Melese, Tewodros Seyoum.

**Project administration:** Wubedle Zelalem Temesgan, Mastewal Belayneh Aklil, Henok Solomon Yacob, Esubalew Tsega Mekonnen, Elias Derso Tegegne, Esubalew Binega Tesfa, Eshetie Melkie Melese, Tewodros Seyoum.

**Resources:** Wubedle Zelalem Temesgan, Mastewal Belayneh Aklil, Henok Solomon Yacob, Esubalew Tsega Mekonnen, Elias Derso Tegegne, Esubalew Binega Tesfa, Eshetie Melkie Melese, Tewodros Seyoum.

**Software:** Wubedle Zelalem Temesgan, Mastewal Belayneh Aklil, Henok Solomon Yacob, Esubalew Tsega Mekonnen, Elias Derso Tegegne, Esubalew Binega Tesfa, Eshetie Melkie Melese, Tewodros Seyoum.

**Supervision:** Wubedle Zelalem Temesgan, Mastewal Belayneh Aklil, Henok Solomon Yacob, Esubalew Tsega Mekonnen, Elias Derso Tegegne, Esubalew Binega Tesfa, Eshetie Melkie Melese, Tewodros Seyoum.

**Validation:** Wubedle Zelalem Temesgan, Mastewal Belayneh Aklil, Henok Solomon Yacob, Esubalew Tsega Mekonnen, Elias Derso Tegegne, Esubalew Binega Tesfa, Eshetie Melkie Melese, Tewodros Seyoum.

**Visualization:** Wubedle Zelalem Temesgan, Mastewal Belayneh Aklil, Henok Solomon Yacob, Esubalew Tsega Mekonnen, Elias Derso Tegegne, Esubalew Binega Tesfa, Eshetie Melkie Melese, Tewodros Seyoum.

**Writing – original draft:** Wubedle Zelalem Temesgan, Mastewal Belayneh Aklil, Henok Solomon Yacob, Esubalew Tsega Mekonnen, Elias Derso Tegegne, Esubalew Binega Tesfa, Eshetie Melkie Melese, Tewodros Seyoum.

**Writing – review & editing:** Wubedle Zelalem Temesgan, Mastewal Belayneh Aklil, Henok Solomon Yacob, Esubalew Tsega Mekonnen, Elias Derso Tegegne, Esubalew Binega Tesfa, Eshetie Melkie Melese, Tewodros Seyoum.

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
