## [Decision Letter · Decision Letter 0]

25 Jan 2022

PONE-D-21-40668Adherence with COVID-19 preventive practice and associated factors among pregnant women in Gondar city, northwest Ethiopia, 2021: Community based cross -sectional studyPLOS ONE

Dear Dr. Temesgan,

Thank you for submitting your manuscript to PLOS ONE. After careful consideration, we feel that it has merit but does not fully meet PLOS ONE’s publication criteria as it currently stands. Therefore, we invite you to submit a revised version of the manuscript that addresses the points raised during the review process.

We look forward to receiving your revised manuscript.

Kind regards,

Sanjay Kumar Singh Patel, Ph.D.

Academic Editor

PLOS ONE

Journal Requirements:

 [No fund was received for this work]. 

3. PLOS requires an ORCID iD for the corresponding author in Editorial Manager on papers submitted after December 6th, 2016. Please ensure that you have an ORCID iD and that it is validated in Editorial Manager. To do this, go to ‘Update my Information’ (in the upper left-hand corner of the main menu), and click on the Fetch/Validate link next to the ORCID field. This will take you to the ORCID site and allow you to create a new iD or authenticate a pre-existing iD in Editorial Manager. Please see the following video for instructions on linking an ORCID iD to your Editorial Manager account: https://www.youtube.com/watch?v=_xcclfuvtxQ.

4. PLOS defines a study's minimal data set as the underlying data used to reach the conclusions drawn in the manuscript and any additional data required to replicate the reported study findings in their entirety. All PLOS journals require that the minimal data set be made fully available. For more information about our data policy, please see http://journals.plos.org/plosone/s/data-availability.

Reviewers' comments:

Reviewer's Responses to Questions

**Comments to the Author**

1. Is the manuscript technically sound, and do the data support the conclusions?

Reviewer #1: Yes

Reviewer #2: Yes

Reviewer #3: Yes

2. Has the statistical analysis been performed appropriately and rigorously? 

Reviewer #1: Yes

Reviewer #2: Yes

Reviewer #3: Yes

3. Have the authors made all data underlying the findings in their manuscript fully available?

Reviewer #1: Yes

Reviewer #2: Yes

Reviewer #3: Yes

4. Is the manuscript presented in an intelligible fashion and written in standard English?

Reviewer #1: Yes

Reviewer #2: Yes

Reviewer #3: Yes

5. Review Comments to the Author

Reviewer #1: I think that it is very important to study the adherence to the preventive protocols, mainly for pregnants in Africa.

I would recommend to correctly call a model with only one explanatory variable as simple logistic regression model. A model with more than one explanatory variable is a multiple logistic regression model. It is wrong to call as bivariable (or bivariate) and multivariable or multivariable.

I proposed few minor corrections in the .pdf appended file.

Reviewer #2: In this paper entitled "Adherence with COVID-19 preventive practice and associated factors among pregnant women in Gondar city, northwest Ethiopia, 2021: Community-based cross-sectional study", the authors investigated the obedience with COVID-19 preventive practices and associated factors among pregnant women. This study is unique because it selected 678 pregnant women during the community-based cross-sectional survey and analyzed the data with rigorous statistics. Moreover, pregnant women are at risk during the pandemic due to physiological reasons. Also, pregnant women are generally excluded from any COVID research and not given vaccines. So this study provided a glimpse of how COVID-19 preventive practices are practiced by pregnant women in Gondar City, North Ethiopia. The manuscript is fascinating and easy to understand. However, there are a few problems in the manuscript.

Minor Comments:

1) A cross-sectional survey of 678 pregnant women in Gondar city can't corroborate the practice practiced by Ethiopia's whole pregnant women population. Mention this in the limitation of the study section.

2) Introduction: The authors should provide information like mortality rate, incubation, various initial prevention approaches, and diet and natural biomolecules for improving immunity and health (doi: 10.1007/s12088-020-00908-0).

3) In the introduction, minor information on the variants of COVID-19 and their future challenges can be included i.e. doi: 10.1007/s15010-021-01734-2.

3) At least one additional Figure (illustration) may be provided to highlight this study's summary or prospect.

Reviewer #3: The manuscript is well written, and it can be accepted after the minor revision. Please find my comments below.

1. Please provide some information on the effect of diet to boost the immunity of pregnant woman towards the treatment of Covid-19 and their variant in the discussion section and the following paper may be followed and cited; Indian Journal of Microbiology, 2020, volume 60, pages 420–429.

2. I would recommend few more figure to describe your explanations instead of tables.

---

## [Author Response · Author response to Decision Letter 0]

31 Jan 2022

Response to Reviewers Date: 31/1/2022

 PLOSS ONE

 Manuscript number: PONE-D- 21-40668

We would like to say thank you for giving us the opportunity to submit a revised draft of our manuscript titled “Adherence with COVID-19 preventive practice and associated factors among pregnant women in Gondar city, northwest Ethiopia, 2021: Community-based cross-sectional study”. We appreciate the time and effort that you have dedicated to provide your valuable feedback on our manuscript. We are grateful to the entire editorial team and peer reviewers for their insightful comments on our paper. We have been able to incorporate changes to reflect most of the suggestions provided by the reviewer. 

Here is a point-by-point response to specific comments and concerns.

Academic Editor’s comments and concerns 

Response: Thank you, dear editor, we kindly accepted your request and we did that in the revised manuscript.

2. If you did not receive any funding for this study, please state: “The authors received no specific funding for this work.”

Response: Thank you, dear editor, we kindly accepted your request and we have corrected in the revised manuscript as “the authors received no specific funding for this work”. Please see the tack change in the revised manuscript.

3. PLOS requires an ORCID iD for the corresponding author in Editorial Manager on papers submitted after December 6th, 2016. Please ensure that you have an ORCID iD and that it is validated in Editorial Manager.

Response: Thank you for pointing this out, I have already created an ORCID id

4. PLOS defines a study's minimal data set as the underlying data used to reach the conclusions drawn in the manuscript and any additional data required to replicate the reported study findings in their entirety. All PLOS journals require that the minimal data set be made fully available

Response: Thank you very much. We have uploaded the minimal data set as supporting information when we submitted the manuscript. If it is necessary, we have uploaded it again as S1 English version questionnaire and S1 Dataset. 

5. Please review your reference list to ensure that it is complete and correct. If you have cited papers that have been retracted, please include the rationale for doing so in the manuscript text, or remove these references and replace them with relevant current references.

Response: Thank you, dear editor, we kindly accepted your request and we have checked all listed references and all are complete, correct and there is no retracted paper. 

Reviewer #1

1.I would recommend to correctly call a model with only one explanatory variable as simple logistic regression model. A model with more than one explanatory variable is a multiple logistic regression model. It is wrong to call as bivariable (or bivariate) and multivariable or multivariable.

Response: Thank you, dear reviewer, for your thoughtful concern. As we tried to review evidence, a multiple or multivariable logistic regression model is a model with more than one explanatory variable with a single binary outcome and we can use interchangeably. However, multivariate implies a statistical analysis with multiple outcomes. Please kindly visit the following articles:

1: Distinction between Two Statistical Terms: Multivariable and Multivariate Logistic Regression: doi:10.1093/ntr/ntaa055.

2: Multivariable Analysis in Cerebrovascular Research: Practical Notes for the Clinician: DOI: 10.1159/000345491

2.I proposed few minor corrections in the .pdf appended file.

Response: Thank you, as per your kind recommendation, we tried to correct in the revised manuscript. Please kindly see the track changes.

Reviewer#2

MinorComments:

1) A cross-sectional survey of 678 pregnant women in Gondar city can't corroborate the practice practiced by Ethiopia's whole pregnant women population. Mention this in the limitation of the studysection.

Response: Thank you, dear reviewer, for your thoughtful concern. We knew that the limitations of the study are those characteristics of design or methodology that impacted or influenced the application or interpretation of the results of the study. Our study finding is generalized to the source population (all pregnant women in Gondar city). But, it is not generalized to Ethiopia's whole pregnant women population. So this not our study limitation.

2) Introduction: The authors should provide information like mortality rate, incubation, various initial prevention approaches, and diet and natural biomolecules for improving immunity and health (doi: 10.1007/s12088-020-00908-0).

Response: The authors would like to thank you for your constructive and helpful feedback. Comment accepted and corrected accordingly. Please kindly see the revised version.

3) In the introduction, minor information on the variants of COVID-19 and their future challenges can be included i.e. doi: 10.1007/s15010-021-01734-2.

Response: Thank you for your constructive and helpful feedback. Comment accepted and corrected accordingly. please kindly see the track change.

3) At least one additional Figure (illustration) may be provided to highlight this study's summary or prospect.

Response: Thank you for your suggestion. We have agreed with your suggestion and corrected in the revised manuscript. Please kindly see the revised manuscript.

 Reviewer #3: 

1. Please provide some information on the effect of diet to boost the immunity of pregnant woman towards the treatment of Covid-19 and their variant in the discussion section and the following paper may be followed and cited; Indian Journal of Microbiology, 2020, volume 60, pages 420–429.

Response: Thank you for your constructive and helpful feedback. Comment accepted and we incorporate it in the introduction section by considering it is the ideal section . Please kindly see the track change.

2. I would recommend few more figure to describe your explanations instead of tables.

Response: Thank you, dear reviewer, we have amended it accordingly. Please kindly see the revised manuscript.

 Best regards!

 Wubedle Zelalem Temesgan

---

## [Editor Report · Decision Letter 1]

7 Feb 2022

Adherence to COVID-19 preventive practice and associated factors among pregnant women in Gondar city, northwest Ethiopia, 2021: Community-based cross-sectional study

PONE-D-21-40668R1

Dear Dr. Temesgan,

We’re pleased to inform you that your manuscript has been judged scientifically suitable for publication and will be formally accepted for publication once it meets all outstanding technical requirements.

Kind regards,

Sanjay Kumar Singh Patel, Ph.D.

Academic Editor

PLOS ONE

---

## [Editor Report · Acceptance letter]

11 Feb 2022

PONE-D-21-40668R1 

Adherence to COVID-19 preventive practice and associated factors among pregnant women in Gondar city, northwest Ethiopia, 2021: Community-based cross-sectional study 

Dear Dr. Temesgan:

I'm pleased to inform you that your manuscript has been deemed suitable for publication in PLOS ONE. Congratulations! Your manuscript is now with our production department. 

Kind regards, 

on behalf of

Dr. Sanjay Kumar Singh Patel 

Academic Editor

PLOS ONE